# AMA: An Ageing Task Migration Aware for High-Performance Computing

**Emmanuel Ofori-Attah and Michael Opoku Agyeman ***

Faculty of Arts, Science and Technology, University of Northampton, Northampton NN1 5PH, UK;
emmanuel.ofori-attah@northampton.ac.uk
* Correspondence: michael.opokuagyeman@northampton.ac.uk

**Abstract:** The dark-silicon challenge poses a design problem for future many-core systems. As a result of this, several techniques have been introduced to improve the number of processing elements that can be powered on. One of the techniques employed by many is Task Migration. In this paper, an Ageing Task Migration Aware for High-Performance Computing (AMA) is proposed to improve the lifetime of nodes. The proposed method determines which clusters applications are mapped to and migrates high-demand tasks amongst nodes to improve the lifetime at every epoch. Experimental results show that the proposed method outperforms state-of-the-art techniques by more than 10%.

**Keywords:** dark silicon; power management; run-time mapping; many-core systems; dynamic voltage frequency scaling (DVFS); task migration; power states; dark core; high computing





## 1. Introduction

Rapid advancement in technology has led to an exponential growth in the number of resources available on a single chip leading to the Many-Core Network-on-Chip (MCNoC) era. With this development, thousands of reduced-sized transistors can be integrated to improve chip performance. Unfortunately, transistor size reduction has led to thermal issues such as high-power density, high chip temperature, and permanent faults caused by leakage power and enforces the dark-silicon phenomenon. The dark-silicon is a phenomenon where more than 30% of chips are predicted to be "dark" or "dumso" at 8 nm [1], limiting full utilization of the abundant of nodes available for maximum performance. Permanent faults cause chip damage and increase the Mean Time To Failure (MTTF) of MCNoCs therefore efficient techniques are required. To prevent chip damage, many-core/multi-core chips have to operate under a fixed budget technique such as Thermal Design Power (TDP) or a fixed per-core power budget such as Thermal Safe Power (TSP) which can also lead to thermal violations [2–5].

In these power budgeting techniques, Dynamic Thermal Management (DTM) techniques such as Task Migration (TM), Dynamic Voltage Frequency Scaling (DVFS), Task Migration (TM), Power Gating (PG), and Near-Threshold Computing (NTC) are used to control the frequency of the chip and to migrate applications or tasks. TDP can be optimistic or pessimistic when implemented causing unnecessary triggering of DTM techniques. TSP is more favoured in recent many-core blueprints because it considers the position of active and dark nodes. Unfortunately, these techniques present challenges. DVFS reduces the temperature of the chip by reducing the frequency which in turn reduces the performance of the chip. On the other hand, TM [6,7] migrates tasks from active nodes to dumso nodes to improve the chip temperature performance. Consequently, in the dark-silicon era, different placements of dark nodes can lead to different thermal profiles of the many-core system.

Unfortunately, this incurs a performance overhead because of memory data transfer and high wake-up latency. Refs. [8–10] proposed a fixed wake-threshold but this may not apply to a many-core system with stochastic applications. Another widely used

technique concerns the number of dark nodes allocated to a specific application. Active nodes can function at higher frequencies when placed near dumso nodes. However, this must be done strategically or there would be insufficient nodes available for the mapping of new applications which can cause delays and performance degradation. Another thing to consider when mapping an application is which node to migrate to. Mapping nodes non-contiguously can lead to communication latency between tasks that require inter-communication.

Other existing literature [11–15] adopts the option of reducing power consumption in Network-on-Chip (NoC). NoC consists of a router and links. The routers contain components such as the arbiter, crossbars, input and output ports, and routers. These components consume about 64% of the NoC's leakage power. Particularly, this is the case for the buffers and the crossbars in the routers since they are identified as the most consuming components in the NoC [16,17].

One technique is the employment of bufferless routers. Ref. [18] propose a heterogeneous NoC architecture comprised of buffered and bufferless routers that reduce the power consumption by 42% in comparison to a buffered router. Unfortunately, this can lead to network congestion. The bufferless router consists of only one local output port for ejection, which can cause several flits to compete for that output port.

Another technique is to reduce power consumption in the cache. Due to their size, LLCs have reportedly been identified to consume the majority of cache leakage power. Several methods have been put up to increase power efficiency. However, these methods can be divided into two groups: cache performance and hybrid designs. The hybrid designs employ Spin-transfer torque magnetic random-access memory (STT-RAM) and static random access memory (SRAM) [19–21]. As an alternative, idle parts of the cache are disabled using power-gating techniques when there is little workload. However, turning off inactive portions of a cache may result in performance costs that increase the amount of power being used.

In [22], less frequently used banks are switched off and their requests are sent to neighbouring requests to ensure that power-gating techniques do not contribute a substantial threat to the performance. Moreover, several design aspects can be employed to improve the dark area on the chips. To address the aforementioned problems, we propose the following novel contributions:

- A dynamic mapping approach to improve the ageing of many-core nodes. The proposed framework is built on our previous contribution ABENA [23];
- The proposed architecture migrates tasks between two nodes depending on the MTTF after an epoch and also during an epoch;
- DVFS and Task Migration are used to reduce the frequency to maintain the temperature of nodes under the temperature constraint.

The paper is organized as follows: Section 2 briefly discusses related work. Section 3 presents the problem statement and in Section 4, the experimental results are presented. Section 5 presents the summary and finally, Section 6 concludes the paper and discusses future work.

## 2. Related Work

Considering the issues surrounding dark silicon, existing work addresses this problem from architectural heterogeneity, on-chip components such as the NoC, Cache, and Core, and Application Mapping techniques. Unfortunately, many existing works do not consider the dark-silicon constraints or process variations and the impact of the ageing of nodes. High temperature accelerates ageing mechanisms such as Electromigration (EM), Negative Bias Temperature Instability (NBTI), time-dependent dielectric breakdown (TDDB), etc. [24,25], leading to a shorter life span for nodes. The lifetime reliability of a node can be reduced by more than 50% by a 10% increase in temperature. Additionally, according to [26], process variation causes nodes to age faster and slower, which affects the lifetime of the chip.

The following work [27–30] tackles the thermal constraints by employing DVFS. However, DFVS alone cannot improve the power and ageing performance of a chip. Ageing depends on several factors such as supply voltage, frequency, stress time, and process variation. Isci et al. [30] introduced a dark-silicon framework that utilises DVFS and dumso node placement to improve the reliability and performance of the chip. Unfortunately, the proposed framework is not scalable for large networks. Additionally, they employ TDP which can be either too pessimistic or optimistic. On the other hand, refs. [31–33] propose online decision-making approaches to overcome the stochastic nature of applications. Unfortunately, due to the large uncertainty of program execution, performance can be degraded.

Haghbayan et al. [34] proposed the Reliability-aware Mapping framework for dark-silicon applications but it does not consider process variation. Alternatively, Rathore et al. [35] proposed the Longevity Framework (LF) to improve the lifetime of nodes. The longevity framework considers process variation in the decision-making of assigning applications to cores. The proposed methodology groups several cores into blocks and then forms a cluster around several blocks. Each cluster consists of dark nodes. Gnad et al. [36] also proposed a Hayat, a dark-silicon framework for ageing deceleration and balancing. They consider process variation in their architecture.

Similarly, Ofori-Attah et al. [23] proposed ABENA. In ABENA, applications are mapped to Nodes with high MTTF after every epoch. In ABENA, NANA is used to form clusters between two nodes. This is to ensure that there is at least one dumso node next to an active node. In the Longevity framework [35], the position of dark nodes is not strategically positioned and controlled. Additionally, they do not consider task migration during an Epoch.

Alternatively, Mohammed et al. proposed dynamic thermal-aware performance optimization (DTaPO), an algorithm that migrates tasks from active nodes to dark nodes. Similarly to ABENA, in DTaPO, dark nodes are not completely power-gated. When the tasks are migrated from the active node, they enter into a low-power state and retain the L2 cache content. This is to allow the new active nodes to access the new dark nodes for information reducing access to the main memory. Clusters are formed between nodes that share the same Last Level Cache (LLC). Similar to ABENA and the Longevity framework, DVFS is used to reduce power consumption. Unfortunately, DTaPo does not consider the MTTF of nodes in its implementation and does not consider periodic workloads.

Wen et al. [37] proposed a dynamic task migration scheme that employs three migration modes. The migration modes are selected based on application characteristics. The first mode is Square Swap Migration where an application is mapped to half of a region. When a hotspot is detected, the entire application is migrated to the next half of the region. The second mode is confined local coolest migration, where an individual task is migrated to a dark core in the same region. The third migration is confined neighbour migration, where a task can be migrated to its neighbouring core which can either be a dark core or an active core if it is below the temperature threshold. Unfortunately, the proposed approach does not consider process variation and ageing. Compared to existing dark-silicon patterning approaches, the proposed scheme considers several factors: process variation, ageing, and the allocation of dark nodes to a specific region.

## 3. System Model and Problem Formulation

### 3.1. Many-Core Platform Model

The many-core system is an $N \times N$ grid of MCA tiles. In the proposed framework, clusters are formed by specifying the number of nodes available and the type of mesh required. Additionally, the number of nodes allocated to each application is dependent on the application with the highest number. This is to ensure that clusters are formed of equal sizes. For example, in a system with 64 nodes where the application with the largest number of tasks is 8, 4 clusters of 16 will be formed. By doubling the nodes in that cluster,

we ensure that in the cluster there are 8 active nodes and 8 dumso nodes. This is because ABENA employs the 50% dumso rule for dark silicon.

Each cluster is allocated a module to monitor the temperature and lifetime of nodes and to utilise DTM techniques to the nodes accordingly. Each module then returns the average temperature and lifetime of its cluster to a centralised resource manager. Based on the information collected, applications are assigned to the right cluster. Additionally, each node has per-core DVFS capability. Each tile consists of a core, a memory (private L1 and L2 caches), and NoC. Unlike [35], the dark nodes enter into a low-power state to allow the active nodes to access the L2 cache contents. Each node has a coordinate of $x$, $y$. The temperature and MTTF of each node are sent to a global manager. Applications are allocated by a resource manager.

### 3.2. Problem Statement

It is important to understand the difference between MTTF and ageing. MTTF is a value generated under existing conditions. Therefore, the MTTF depends on several factors: temperature and the number of active and dumso nodes. Additionally, it also depends on the current applications being executed. Different applications have different characteristics which can change the MTTF of a node. Furthermore, the tasks from a particular application can also have an impact on the MTTF of every single node.

On the other hand, the ageing of a system is measured over a period of time. Exiting work compares the temperature of nodes after an epoch and assigns applications after an Epoch. As a result of the stochastic nature of applications, the MTTF during an epoch can change.

### 3.3. ABENA 2.0 Overview

At any given time, a total $n$ of applications arrive at the system. Each application consists of several $T$ threads. The objective is to map application tasks to nodes whilst satisfying the thermal constraints and performance. ABENA has two different modes.

ABENA 2.0 inherits the majority of the algorithm structures from ABENA 1.0 with an additional clustering feature. In ABENA 2.0 clustering is formed in two different ways. The clustering can be performed by specifying the type of mesh required or by forming clusters based on the LLC that the nodes share as depicted by [7]. Similar to ABENA 1.0, the first cluster to map the first application is chosen by calculating the $MTTF$ of all the clusters. Then, the cluster with the highest $MTTF$ is chosen. This process is repeated until the last step. If two clusters return the same $MTTF$, the $C$ with the lowest $T$ is chosen. ABENA 2.0 also inherits Neighbour-Temperature-Aware Node-Allocation (NANA) to form mini clusters between two neighbouring nodes. NANA compares the MTTF of the two neighbouring nodes and maps the task to the nodes that return the highest MTTF. DVFS is then used to reduce the voltage frequency by 10%. This 10% is not a special number. This number is chosen because increasing and decreasing the VF by 1% does not have a significant impact on the temperature. Algorithm 1 depicts ABENA 2.0.

In [4], the execution of blackscholes shows how threads are executed. In Phase 1, only the parent threads are executed. In Phase 2, the slave threads are executed and in Phase 3, the master threads are executed. By using this analysis, they proposed the Transient-Temperature Based Safe Power Budgeting (TSP) to fully exploit TSP budgeting. We derived the motivation from [4]. Based on these conclusions, we propose An Ageing Task Migration Aware for High-Performance Computing(AMA) to improve ABENA 2.0.

---

**Algorithm 1:** ABENA Periodic [23]

---

**Data: Data**: $C_1...C_N$

**Result:** Low to High Clusters

1  $N$ : *Nodes* $C$ : *Clusters* ;

2  $C_{SET}$ : *ClusterSet* ;

3  $NC_1...NC_1$ : Mini Node Clusters ;

4  *ReturnAverage* : *ReturnAverageoftheCluster* ;

5  $MTTF^o$ : *MeanTimeToFailure* ;

6  $T_{TRS}$ : *TemperatureThreshold* ;

7  *Temp* : *Temperature* ;

8  *NFrequency* : *Frequency* ;

9  **for** $C \in C_{SET}$ **do**

10  $\quad$ $ReturnAverage \longleftarrow N_K \div (\sum N_1^{MTTF} \times N_K^{MTTF})$;

11  **end**

12  $FirstCluster \longleftarrow CwiththeHighestMTTFAverage$ ;

13  * Initial Mapping * Select 50% of the nodes with the lowest temperature nodes from each cluster ;

14  **if** $N_{1Temp^o} > T_{TRS}$ : **then**

15  $\quad$ $N_k \Longleftarrow ApplyDVFS \; N_{Frequency}$ - 10% ;

16  **end**

---

### AMA: Ageing Task Migration Aware for High-Performance Computing

During periodic workloads, some applications spawn threads that consume more power and function more than the rest of the tasks. A conducted experiment using four different Fast Fourier Transform (fft) applications shows the first task of each application has a lower idle time compared with the rest of the tasks. Table 1 depicts the simulation parameters of a conducted study on the impact that benchmarks have on the MTTF and Temperature. For this experiment, fft and blackscholes were chosen as the benchmark.

**Table 1.** Summary of Simulation Parameters.

| Parameter | Value |
|---|---|
| Application | fft |
| Temperature threshold | 90° |
| Failure Mechanism | TDDB |
| L1 Instruction Cache | 32 kB, 4-way set-associative, 64B Block Size |
| L2 Cache | 51 kB, 4-way set-associative, 64B Block Size |
| L3 Cache | 8129, 23 Block Size |
| Technology | 22 nm |
| Network size | $8 \times 8$ NoC |
| Algorithm | ABENA-P |
| DVFS | No |

Table 2 shows the simulation results. It can be deduced that Task 1 from each application is utilised more. The difference is 40%. There is a 2% decrease of MTTF on all the nodes Task 1 is assigned to. From the results, we also notice that the floor plan of the architecture impacts the MTTF and the nodes. To obtain an unbiased setup, we conducted the set for each core to experience the same effects regardless of its position. This shows the utilisation of the node with the first task is influenced by the task. Based on this result and conclusion, we present AMA to improve this utilisation time.

**Table 2.** Summary of Simulation Parameters.

| Parameter | Utilisation (%) | Temperature (°) | Av. MTTF | Highest MTTF |
|---|---|---|---|---|
| Task 1 | 52–53 | 74–77° | 469.225 | 542 |
| Tasks 2–8 | 93–95 | 71–77° | 463.53 | 492 |

Algorithm 2 depicts the algorithm AMA. In Lines 2–3, for each mini cluster, AMA checks whether the high-dependent task has already been used on both nodes. If the task has not been used by both nodes, the highly dependent task is assigned to the node that it has not been assigned to yet. If the task has already used both nodes in that cluster, the task is moved to the next neighbouring cluster in a round-robin style. In this case, we share the utilisation and power consumption amongst the cluster. Figure 1 shows the overall architecture of the considered system. Figure 1a shows Epoch 1. At Epoch 1, NANA is used to select which node has the lowest MTTF. At Epoch 2, NANA is used again to choose the next node. In this epoch, the second node is used to assign the task. In Epoch 3, as Task 1 has already been used in the first cluster, at Epoch 3, the task is moved to the next mini-cluster.

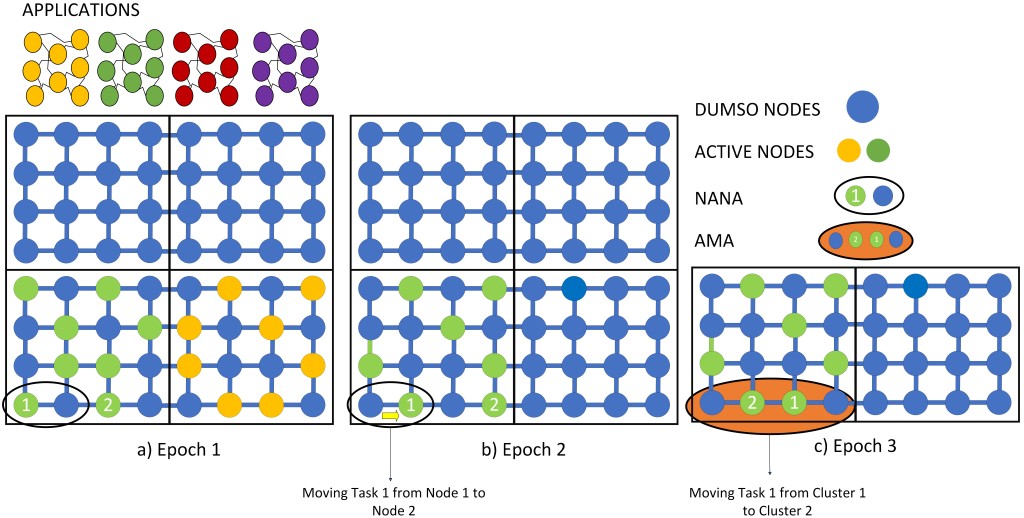

**Figure 1.** Proposed Architecture.

---

**Algorithm 2:** AMA Algorithm

**Data:** $NC_1...C_N$ Node Cluster
1 **for** $NC_1 \in \mathcal{C}_N$ **do**
2     **if** $T_i \implies NC_1$ **then**
3        $T_i = NC_2$
4     **end**
5 **end**

---

## 4. Experimental Evaluation

This section presents the experimental setup, results, and discussions of the proposed work.

### 4.1. Experimental Setup

Experiments are performed on LifeSim, an extended version of Sniper [38] interfaced by McPat [39] for power details, HotSpot [40] for thermal modelling and Ramp [41] for lifetime modelling. Table 3 depicts the parameters for the simulated configuration. Splash-2 Benchmark applications are used. The experimental setup is shown in Figure 2. McPat outputs the power details of the entire simulation. The power details are then passed to the

HotSpot. The HotSpot uses the power output to generate a temperature. The temperature traces are then fed to RAMP. RAMP generates the ageing and MTTF of the nodes based on the failure mechanism. ABENA uses the temperature generated, ageing, and the MTTF to make the mapping decisions.

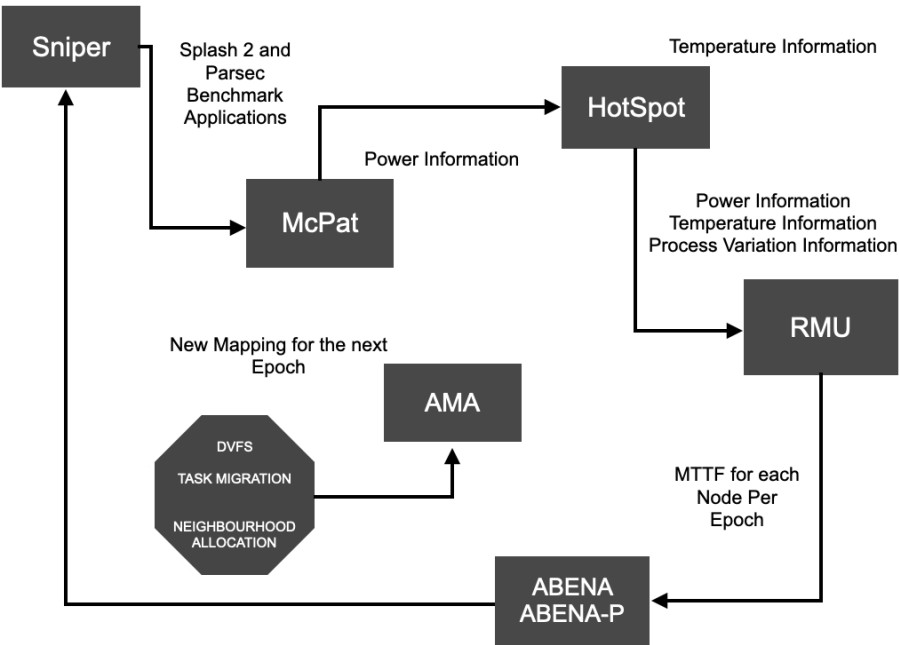

**Figure 2.** The environmental setup of the proposed method.

**Table 3.** Summary of Simulation Parameters.

| Parameter | Value |
| --- | --- |
| Nodes | 64 |
| Core Frequency | 1 GHZ |
| Core Size | ×5.71 |
| L1 Data Cache | 32 kB, 4-way set-associative, 64B Block Size |
| L1 Instruction Cache | 32 kB, 4-way set-associative, 64B Block Size |
| L2 Cache | 512 kB, 4-way set-associative, 64B Block Size |
| L3 Cache | 8129, 23 Block Size |
| Applications | Radix, oceans, fft, barnes |
| Technology | 22 nm |
| Network size | 8 × 8 NoC |
| Temperature threshold | 90° |
| Failure Mechanism | TDDB |
| Algorithm | AMA |

For the hardware platform, we simulate a 64-node many-core system in an 8 × 8 grid interconnected by an NoC. The traditional NY router is used with the NoC. Each node has a per-core frequency range of 1 Ghz with steps of 100 MHz. Consequently, we implemented 6 different VF levels, as shown in Table 4. The technology implemented is 22 nm. The node size is 0.7 mm × 0.8 mm and comprises of Nahalem core. Each node consists of L1 cache (256 kB), and L2 cache (512). The nominal frequency is 1 GHz.

**Table 4.** DVFS VF Levels.

| Voltage (in V) | Frequency (in MHz) |
| --- | --- |
| 1 | 1000 |
| 0.9 | 900 |
| 0.8 | 800 |
| 0.7 | 700 |
| 0.6 | 600 |
| 0.5 | 500 |

For accurate results, we modified LifeSim to support dark silicon with the experiment conducted 10 times. Context switching in this simulation is not measured because Snipersim only simulates user-space and does not simulate the operating system and the context switched managed by it [35]. In theory, context switching usually takes 5–7 ns, but in this case study, intervention occurs after every epoch and because we use non-preemptive scheduling in this study, the time overhead and performance are very minimal [42].

The experiments are conducted with a workload comprised of SPLASH-2 [43] and PARSEC benchmarks suits. We consider periodic workloads because ageing is measured over a long period of time. We also consider a mix of applications from these two benchmarks to allow different application characteristics to be examined.

The temperature cap for the experiment is 90°. Similar to LF [34], we chose an epoch as one month for evaluation. This is because a long epoch would have caused too much degradation by the time there is an intervention and choosing too small of an epoch does not allow enough time for the applications to stress the nodes. The failure mechanism chosen is TDDB. However, the proposed mechanism can also be applied to other failure mechanisms.

We compared AMA with two state-of-the-art approaches—ABENA 2.0 and DTaPO [7]. For clarification purposes, we refer to ABENA 3.0 as AMA. The evaluated parameters are MTTF and Utilization. In [7], the simulation results showed that, at 85°, there are 0 migrations and 0 average of DVFS. For this purpose, we also adopt 90° as the temperature cap.

### 4.2. Results

This section explains the results of the conducted study.

### 4.3. Average Comparison of MTTF

Figure 3 shows the normalised average of the MTTF for ABENA, AMA, and DTaPO. The data show that DTaPO has the highest average. This is because, DTaPO does not consider MTTF and as a result, the same node is selected for every single epoch. This is evident in the highest average node result. It shows DTaPO has the highest node. The reason for this is because that node is never used. DTaPO only utilises other nodes when the temperature cap is exceeded. In this experiment, the temperature cap was not violated leading to the same nodes being used every epoch. This can be concluded from the lowest average node. In the lowest average node, DTaPO has the lowest node because the same node is being used.

Additionally, AMA also outperforms ABENA 2.0. The average of AMA is higher than ABENA and it also has a higher MTTF node and lower MTTF node. This is because of the reduction of VF of the low-utilisation nodes. There is an increase in the MTTF as they are not utilised as much as the nodes with heavy-demand tasks.

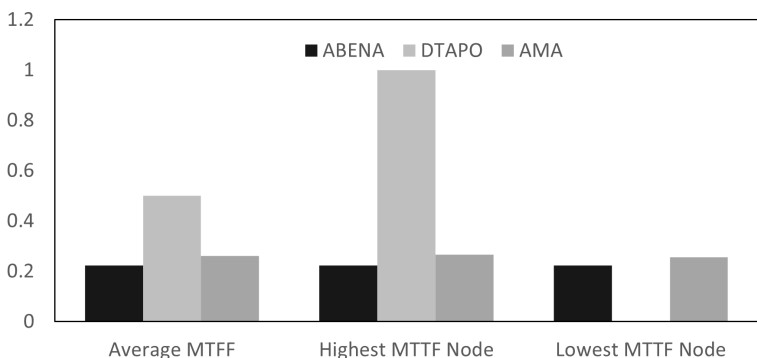

**Figure 3.** MTTF Comparison between ABENA, AMA and DTaPO.

### 4.4. Average Comparison of Temperature

Figure 4 shows the normalised average of the Temperature for ABENA, AMA, and DTaPO. The data show that all three algorithms have a similar temperature. This is because all three algorithms adopt one-on and one-off chessboard patterns. It is important to note that one of the main factors that contribute to MTTF is the temperature and power of a node. The power of a node contributes to the temperature of the node as well as its position. A node with a very high temperature surrounded by the other active nodes causes the MTTF to be high, hence the employment of the one-off and one-off pattern by these three algorithms. In this experiment, both ABENA and AMA outperform DTaPO when comparing the highest MTTF node. This is because the DVFS of the Nodes in DTaPO are never reduced unless they violate the temperature cap. The advantage of this procedure is that tasks can meet their deadline and finish quicker. Unfortunately, in the long run, as shown, for MTTF, the nodes are high.

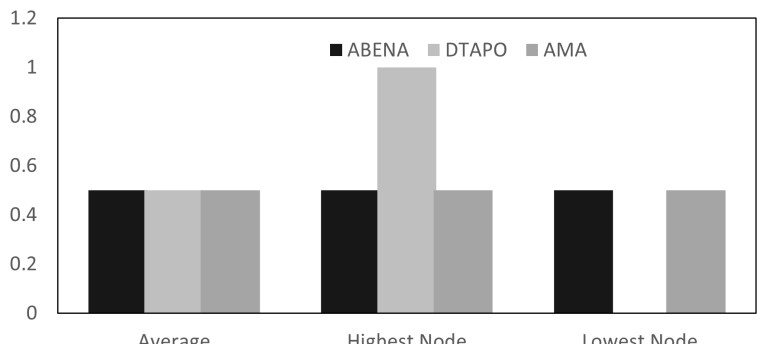

**Figure 4.** Temperature Comparison between ABENA, AMA and DTaPO.

### 4.5. Comparison of Utilisation

Figure 5 shows the normalised average of the utilisation of the nodes conducted from the experiment. It can be deduced from the results presented that AMA outperforms both ABENA 2.0 and DTaPO. Although there are similarities when comparing the overall average and the highest node, the lowest node comparisons prove that AMA is better. In the lowest node comparison, we can see that the DTaPO overuses its cores. The same node is selected each time for each task every epoch. With ABENA and AMA, different nodes are used. However, AMA swaps the heavy-duty task around to improve the utilisation and thus improve the MTTF.

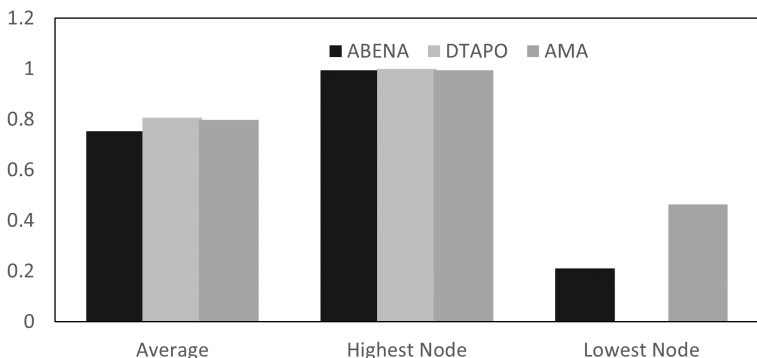

**Figure 5.** Utilisation Comparison between ABENA, AMA and DTaPO.

*4.6. Execution Time*

Figure 6 shows the execution time of the project. It can be deduced that the simulation with DTaPO was completed quicker than the simulation of ABENA and AMA. This is because the changing and switching of tasks amongst different nodes increase the time it takes for tasks to be completed. We can conclude from this result that DTaPO outperforms ABENA and AMA in the execution time at the expense of node degradation.

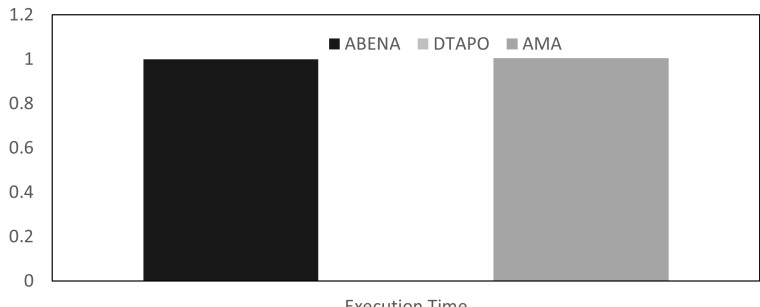

**Figure 6.** Execution Time Comparison between ABENA, AMA, and DTaPO.

**5. Summary**

Simulation results show that AMA improves the utilisation of the nodes. In DTaPO, tasks are only migrated when there is a temperature violation. In ABENA 2.0, a fresh set of nodes is used for each epoch based on the MTTF. Unfortunately, the utilisation of the first task causes its assigned core to have a higher utilisation percentage than the rest. AMA moves the first task amongst the clusters for each epoch and ensures that each node in a mini cluster is assigned the first task to balance the utilisation and improve the MTTF. Figure 6 depicts the utilisation of all three approaches. It is observed that the proposed technique keeps the utilisations close to each other. The difference between AMA and DTaPO is more than 25%, whilst the difference between AMA and ABENA 2.0 more than 5%. However, DTaPO outperforms ABENA and AMA when comparing simulation time by more than 10%.

**6. Conclusions**

In this paper, we presented AMA, an Ageing Task Migration Aware for High-Performance Computing algorithm to improve the lifetime of nodes, temperature, and utilisation. Simulation results show that the proposed algorithm outperforms the existing technique. AMA maps tasks with high utilisation amongst nodes to avoid placing the burden on just a few specific nodes. Additionally, this improves the lifetime of the nodes when the nodes do not exceed the temperature cap. Moreover, the proposed framework outperforms DTaPO by more than 20% in MTFF. AMA also outperforms in terms of utilisation by more than 30%. Although this seems very insignificant, for a periodic workload system where applications are executed (10–20 epochs), the lifetime of each node is extended compared to other

techniques. In future work, we plan to improve the MTTF of each node and also improve the execution time of AMA. Currently, DTaPO executes applications quicker than AMA.

**Author Contributions:** Methodology, Validation, Resources, Writing–original draft preparation, review and editing, E.O.-A., Subversion, writing—review and editing, M.O.A. All authors have read and agreed to the published version of the manuscript.

**Funding:** This research received no external funding.

**Data Availability Statement:** The supporting dataset for this manuscript is available from the University of Northampton Research Explorer at http://doi.org/10.24339/390923d7-fa26-4324-b76d-0a5b9dc80969Í, accessed on 22 February 2023.

**Conflicts of Interest:** The authors declare no conflict of interest.

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
