# Peer review of "AMA: An Ageing Task Migration Aware for High-Performance Computing"

_jlpea, doi:10.3390/jlpea13020036_

Round 1

Reviewer 1 Report

Dear Authors,

Your manuscript is well-organized and your contribution to the state of art is identified, but the clarity in several passages should be improved. Please find some major and minor comments below.

Major comments

You claim that the mean time to failure and aging of nodes improves with respect to other relevant techniques. Although the results are discussed in detail, it is still unclear how the better workload distribution improves the parameters. Please clarify.

Please clarify lines 236-300. You define MTTF_TDDD but then introduce MTTF_EM, please also correct the formula. Are you are meaning to write an equation with proportionality factor? In this case, please replace the infinity sign. ou evaluate MTTF as a percentage, while it is by definition a time interval.  Please correct and improve your statements.

In several sentences you tend to use qualitative statements, such as, for example at line 46, “These components consume a lot of power.” Please make an effort to quantify your statements. In this specific example, which percentage of power is consumed by the components? Also, please revise the full manuscript for qualitative statements.

Minor comments

Line 52, Cache -> cache (use lowercase ‘c’)

Line 106, MTFF, acronym undefined, please define it (or is it a typo-ed MTTF?)

Line 131, Nx x Nx, is the x subscript really needed?

Line 190, Table KB -> kB (the kilo prefix requires lowercase ‘k’)

Line 228, 5-7n -> 5-7ns ?

Line 245, The parameters that the results are conducted on are … -> The evaluated parameters are …

Line 299, The sentence lacks an ending. Please fix it.

Author Response

You claim that the mean time to failure and aging of nodes improves with respect to other relevant techniques. Although the results are discussed in detail, it is still unclear how the better workload distribution improves the parameters. Please clarify. 

  • The mean time to failure improves because the task that consumes more power is passed around amongst the mini clusters. This is because some tasks consume more power and increase the MTTF. By passing these tasks around, we balance the MTFF and the utilization of the nodes. 

Please clarify lines 236-300. You define MTTF_TDDD but then introduce MTTF_EM, please also correct the formula. Are you are meaning to write an equation with proportionality factor? In this case, please replace the infinity sign. ou evaluate MTTF as a percentage, while it is by definition a time interval.  Please correct and improve your statements. 

  • The formula has been removed from the paper as it is not relevant to this paper. 

In several sentences you tend to use qualitative statements, such as, for example at line 46, “These components consume a lot of power.” Please make an effort to quantify your statements. In this specific example, which percentage of power is consumed by the components? Also, please revise the full manuscript for qualitative statements. 

  • NoC consists of a router and links. The routers contain components such as the arbiter, crossbars, input and output ports and routers. These components consume about 64\% of the NoC's leakge power. 

Minor comments 

Line 52, Cache -> cache (use lowercase ‘c’) 

  • The Cache word as been changed to cache 

Line 106, MTFF, acronym undefined, please define it (or is it a typo-ed MTTF?) 

  • The MTFF is a typo. It has been changed to MTTF 

Line 131, Nx x Nx, is the x subscript really needed? 

  • In all of our previous journals, we have used this. Thank you for highlighting this. We have removed this.   

Line 190, Table KB -> kB (the kilo prefix requires lowercase ‘k’) 

  • Changed the prefix to kB as advised.  

Line 228, 5-7n -> 5-7ns ? 

  • Amended as advised. 

Line 245, The parameters that the results are conducted on are … -> The evaluated parameters are  

  • Amended as advised 

Line 299, The sentence lacks an ending. Please fix it. 

  • The sentence has been removed 

Reviewer 2 Report

The manuscript entitled, "AMA: An Ageing Task Migration Aware for High Performance Computing." is thoroghly reviewed and it is technically good. However, I have following comments or suggestions for the further improvements:

1) Write the full form of abbreviations when they appear in the beginning (for example, DTaPO, etc.).

2) There is discontinued sentences in the section 3.2, for example, "Therefore, the MTTF of a core 154 can be modelled as:"..........(please complete it).

3) What is the limitation of the algorithm 2 used in the manuscript? Please explain it and I think, the explanation about the algorithm 2 must be in the section 4. Experimental Evalution, since there are only Algorithm 2 with no explanation.

4) Figures 3 and 4 are not clear, please replace it if possible.

5) Please add the significance of the result in the conclusion part.

6) In the summary section (4.7), the sentence is incomplete. Please complete it and rewrite the appropriate sentence. ( This is very significant and important when running the simulation once but for 299 periodic workload, the............................)

Author Response

The manuscript entitled, "AMA: An Ageing Task Migration Aware for High Performance Computing." is thoroughly reviewed and it is technically good. However, I have following comments or suggestions for the further improvements: 

1) Write the full form of abbreviations when they appear in the beginning (for example, DTaPO, etc.). 

  • The full meanings of all abbreviations have been added to paper.  

2) There is discontinued sentences in the section 3.2, for example, "Therefore, the MTTF of a core 154 can be modelled as:"..........(please complete it). 

  • The statement has been removed from the paper 

3) What is the limitation of the algorithm 2 used in the manuscript? Please explain it and I think, the explanation about the algorithm 2 must be in the section 4. Experimental Evalution, since there are only Algorithm 2 with no explanation. 

  • We have added a few lines to explain the algorithm.  

4) Figures 3 and 4 are not clear, please replace it if possible. 

5) Please add the significance of the result in the conclusion part. 

  • Moreover, the proposed framework outperforms DTaPO by more than 20% in MTFF. AMA also outperforms the Utilisation by more than 30%. 

6) In the summary section (4.7), the sentence is incomplete. Please complete it and rewrite the appropriate sentence. (This is very significant and important when running the simulation once but for 299 periodic workloads, the............................) 

  • The last statement has been removed from the paper. 

Round 2

Reviewer 1 Report

Dear authors,

thank you for revising the manuscript, however in the next revisions I advise to highlight changes in the manuscript, e.g., use another color, to quickly identify them.

Concerning the MTTF, one key point is the relationship between MTTF and temperature, the relationship between dissipated power and temperature, which should be shown explicitly in the paper.

Is the MTTF predicted only by means of simulations? Did you perform any actual measurement? If so, including the experimental results would strengthen your paper.

Please also check the manuscript for typos and missing/incorrect references.

Regards

Author Response

thank you for revising the manuscript, however in the next revisions I advise to highlight changes in the manuscript, e.g., use another color, to quickly identify them.

  • We have used the yellow marker to highlight the changes as adviced.

Concerning the MTTF, one key point is the relationship between MTTF and temperature, the relationship between dissipated power and temperature, which should be shown explicitly in the paper.

  • It is important to note that, one of the main factors that contribute to MTTF is the temperature and power of a node. The power of a node contributes to the temperature of the node as well as the position of the node. A node with a very high temperature surrounded by the other active nodes causes the MTTF to be high hence the employment of the one-off one-off pattern by these three algorithms.

Is the MTTF predicted only by means of simulations? Did you perform any actual measurement? If so, including the experimental results would strengthen your paper.

  • The predictions are only a means of simulations. We used LifeSim, an extended version of Sniper.

 Please also check the manuscript for typos and missing/incorrect references.

  • We have reviewed the paper and corrected the types.